# Testing the Shielding Effect of Intergenerational Contact against Ageism in the Workplace: A Canadian Study

**DOI:** 10.3390/ijerph19084866

**Published:** 2022-04-16

**Authors:** Martine Lagacé, Anna Rosa Donizzetti, Lise Van de Beeck, Caroline D. Bergeron, Philippe Rodrigues-Rouleau, Audrey St-Amour

**Affiliations:** 1Department of Communication, Faculty of Arts, University of Ottawa, Ottawa, ON K1N 6N5, Canada; prodr027@uottawa.ca; 2School of Psychology, Faculty of Social Sciences, University of Ottawa, Ottawa, ON K1N 6N5, Canada; lvand103@uottawa.ca (L.V.d.B.); abouc005@uottawa.ca (A.S.-A.); 3LIFE Research Institute, University of Ottawa, Ottawa, ON K1N 6N5, Canada; caroline@bergeron.ca; 4Department of Humanities, University of Naples Federico II, 80138 Naples, Italy; annarosa.donizzetti@unina.it; 5Center for Population Health and Aging, Texas A&M University, College Station, TX 77843, USA

**Keywords:** older workers, intergroup contacts, knowledge sharing, work engagement, intention to remain in the workplace

## Abstract

Negative outcomes of ageism in the context of the Canadian labor market are well documented. Older workers remain the target of age-based stereotypes and attitudes on the part of employers. This study aims at assessing (1) the extent to which quality and quantity intergroup contacts between younger and older workers as well as knowledge-sharing practices reduce ageist attitudes, in turn (2) how a decrease in ageist attitudes increase the level of workers’ engagement and intentions to remain in the organization. Data were collected from 603 Canadian workers (aged 18 to 68 years old) from private and public organizations using an online survey measuring concepts under study. Results of a path analysis suggest that intergroup contacts and knowledge-sharing practices are associated with positive attitudes about older workers. More so, positive attitudes about older workers generate higher levels of work engagement, which in turn are associated with stronger intentions to remain with the organization. However, positive attitudes about older workers had no effect on intentions to remain in the workplace. Results are discussed in light of the intergroup contact theory.

## 1. Introduction

The COVID-19 pandemic has allowed for greater intergenerational solidarity [1,2]. However, it has also exacerbated expressions of ageism whereby by older adults were often stereotypically portrayed as members of a homogenous and vulnerable group [3,4,5,6,7]. More so, from the very beginning of the crisis, certain countries and states’ public policy decisions implied that COVID-19 was mainly a problem of older adults which justified the need to isolate them from the young [8,9,10]. How will such age-based segmentation play in a post-pandemic time is a question that urgently needs to be addressed, acknowledging that ageism actually stems and thrives from the social separation of the young and old [11]. Ageism relates to stereotypes and discrimination based on age and, according to Butler [12], mainly targets older adults as it “allows the younger generations to see older people as different from themselves, thus they subtly cease to identify with their elders as human beings” (p. 35). Findings from several studies have suggested that ageism is prevalent in different social arenas such as education, healthcare as well as the labor market, which is the focus of this study [13,14,15,16,17,18].

A scoping review conducted by Harris et al. [19] summarized the most common ageist stereotypes conveyed by employers towards older workers: positive stereotypes include perceptions that older workers (defined as workers aged 50 years old or more in this paper) possess soft skills, and are loyal and reliable employees. Notwithstanding these qualities, older workers are negatively portrayed as being less capable of adapting to change—especially to new technologies—less capable of learning, less competent and having decreased performance activity. In other words, the skills of multitasking and fast learning that many employers value in modern work environments are associated mainly with younger workers [20].

Studies have documented older workers’ reactions to such characterizations, and the negative psychosocial outcomes resulting from the perception of being the target of ageist stereotypes by employers. For example, older workers confronted with negative age-based stereotypes have higher levels of stress and lower affective commitment to their organization [21,22]. Along the same lines, in a Canadian study conducted among older health care workers, Lagacé and al. [23] have shown that the perception of ageist attitudes on the part of employers as well as colleagues lead to psychological disengagement from the workplace, which in turn, negatively impacts self-esteem. In a Belgium study, Gaillard and Desmette [24] have also shown that exposure to negative age-based stereotypes reinforce older workers’ intentions to retire early. Finally, ageism in the workplace is associated with a lower quality of life [25].

### 1.1. Beyond Perceptions: Employers’ Ageist Recruiting and Training Practices

Much research on ageism within the context of employment relates to older workers’ perceptions of being the target of ageist stereotypes and to employers subscribing to such stereotypes. However, more and more studies are empirically documenting ageist behaviors—beyond mere perceptions—of employers towards older workers as well as the impact of such behaviors precisely on recruitment and training opportunities. For example, Solem’s study [26], which looked at a large sample of Norwegian employees and managers, shows that although the latter perceive employees above 50 years of age to be performing as well as younger employees, they remain hesitant to call older applicants to job interviews.

Along the same lines, Neumark, Burn and Button [27] designed and implemented a large field experiment study to test how age-based discrimination permeates employment, namely through hiring practices. Based on evidence from more than 40,000 job applications for administrative jobs, submitted by young applicants (aged 29 to 31 years old), middle-aged applicants (aged 49 to 51 years old) or older applicants (aged 64 to 66 years old), they found stronger evidence of discrimination (i.e., call back rate) against applicants near retirement ages than middle-aged applicants. Findings from this study are particularly relevant since they reveal the importance of including and engaging older workers nearing normal retirement age when addressing age-based discrimination.

Training opportunities is another area where ageist practices on the part of employers have been documented. While lifelong learning is one of the means to support an aging workforce and retaining older workers, there is still uncertainty among managers regarding the desirability and feasibility of training older workers. One of the outcomes is that employers generally provide less training to older employees, starting at age 50, than they do to younger employees. Findings from a 2015–2016 study involving nine European countries and 2517 employees and their managers spread over 228 organizations, confirmed that, with regard to training opportunities, people are treated differently based on their age, and that employers provide more training to younger employees than they do for older ones [28]. According to the authors, the reason for such practices relates to prejudice and stereotypes about aging in the workplace, where older workers are judged to be less willing and less able to be trained. Consequently, managers consider the return on investment in training older workers to be too low.

In sum, the manifestations and negative outcomes of ageism in the context of the labor market are now well documented. Findings of studies show that employers and managers still rely on ageist stereotypes (although these are not always explicitly expressed) when it comes to hiring and training decisions and that such decisions put older workers (and older applicants) at a disadvantage in comparison to their younger counterparts. Yet, in the face of global labor shortages—exacerbated by the COVID-19 pandemic [29,30]—there is an urgent need to address ageism as one important barrier in terms of hiring and retaining older workers in the job market. In Canada, for example, recent efforts have been made to promote the labor force participation of older Canadians [31]. Hence, the main goal of the current study is to examine factors that may significantly counter ageism in the Canadian workplace. One of these, as suggested by Burnes and colleagues’ study [32], relates to intergenerational contacts. These researchers conducted a meta-analysis of 63 peer-reviewed studies aiming to assess the efficacy of interventions to reduce ageism. The findings reveal that combined education and intergenerational contacts showed the largest effects in terms of countering ageist attitudes.

To shed further light into the efficacy of intergenerational contacts in reducing ageist stereotypes and discrimination, we first turn to the intergroup contact theory (ICT) and then summarize findings of studies that have relied on ICT in the context of the workplace.

### 1.2. Intergroup Contact Theory

Conceptualized by Allport [33], ICT has been one of the most promising theories in explaining why and how negative stereotypes and discrimination can be reduced. The theory states that in the presence of certain conditions (equal status, common goals, inter-group cooperation and support by social and institutional authorities), contact between members of different social groups significantly decrease prejudice. The first condition relates to how groups expect and perceive equal status in the situation while the second, common goals, is about contact that requires active and goal-oriented effort. Athletic teams are an example of this second condition. Intergroup cooperation relates to the interdependent nature of the efforts needed to achieve the common goals but without intergroup competition. Finally, the fourth condition states that intergroup contacts are more easily accepted when explicitly supported by authorities. Interestingly, Pettigrew and Tropp [34] conducted a meta-analysis of 515 studies that revealed that, even when the four optimal conditions are not met, intergroup contact still reduces prejudice. Exploring the mechanisms through which intergroup contacts have a positive effect on how members of different social groups perceive one another, Swart and colleagues’ longitudinal study [35] shows that these contacts actually reduce intergroup anxiety over time and induce positive affective processes. While intergroup contacts have often been studied as they relate to discrimination based on race, culture and gender, the effect of these contacts were less examined with regard to the issue of ageism, especially in the context of work. For example, Iweins and colleagues [36] measured the outcomes of the quality of contacts between younger and older Belgium workers from the perspective of 129 employees aged less than fifty years old. The findings suggest that age-based intergroup contacts mitigate ageist attitudes toward older workers through a dual-identity process (whereby age-based ingroup–outgroup identity distinction is maintained but is somewhat weakened by a common work identity). Along the same lines, a study conducted by Lagacé and colleagues [37], comprised of 415 Canadian older workers, confirmed that positive intergenerational workplace climate, fostered by a positive perception of quality and quantity intergroup contacts between younger and older workers, had a mitigating impact on perceived ageism. In turn, lower perceptions of ageism were associated with higher levels of satisfaction at work. The study also included a measure of “knowledge sharing between younger and older workers” conceptualized as the behavioral component of a “positive intergenerational workplace climate”. Both of these two variables were strongly associated. While findings of the latter study shed further light on the critical role that a positive intergenerational workplace climate plays on reducing perceptions of ageism, it did not include the perspective of younger workers. More so, the final model of that same study did not focus on outcomes that are critical in the face of current labor shortages in Canada, namely, work engagement and intentions to remain. Canadian employers and managers are facing major barriers relating to work engagement and retention of workers, particularly of older workers [38,39,40].

### 1.3. Study Objectives and Hypotheses

Hence, building on the postulates of ICT as well as the findings from the previous studies stated above, the current study aims at assessing: (a) the impact of age-based intergroup contacts and knowledge-sharing practices on ageism; and (b) the impact of ageism on workers’ engagement and intentions to remain in the organization, from both the perspectives of younger and older Canadian workers.

Hypotheses are as follows:

**Hypothesis** **1.**
*Quantity and quality age-based intergroup contacts are associated with positive attitudes about older workers.*


**Hypothesis** **2.**
*Age-based knowledge-sharing practices are associated with positive attitudes about older workers.*


**Hypothesis** **3.**
*Quantity and quality age-based intergroup contacts are correlated with age-based knowledge-sharing practices.*


**Hypothesis** **4.**
*Positive attitudes about older workers generate higher levels of work engagement.*


**Hypothesis** **5.**
*Positive attitudes about older workers increase intentions to remain in the organization.*


**Hypothesis** **6**. *Higher levels of work engagement are associated with higher level of intentions to remain within the organization.*

## 2. Materials and Methods

### 2.1. Recruitment

After obtaining ethics approval from the first author’s institution, the team of researchers recruited participants through online invitations. Precisely, an email invitation (drafted by researchers) was sent to employers of private and public Canadian organizations from August 2020 to April 2021. The invitation stated the independent and confidential nature of the study and included a link to the online questionnaire created using the survey software platform Qualtrics. Employers were asked to forward the invitation to workers in their organization (as researchers did not have access to workers’ email addresses due to confidentiality reasons). Inclusion criteria was as follows: part-time or full-time workers, aged 18 years or more, able to understand and read French or English. Participants who agreed to take part in the study simply clicked on the link to provide consent and to complete the questionnaire; after completion, answers were automatically sent to researchers.

### 2.2. Participants

A total of 745 participants took part in the study. After screening for missing data, a total of 603 workers (340 women, 263 men) were retained for the final sample. Key demographic characteristics are presented in Table 1. Age range was 18 to 68 years old (*n* = 490 aged from 18 to 49 years old; *n* = 114 aged 50 to 68 years old) with an average age of 37.58 years (SD = 11.53). A majority of participants worked more than 21 h a week (*n* = 546) and a total 335 worked in the Canadian public sector. In addition, 155 participants were in a management position. Finally, 314 participants reported French as their first language, 229 reported English, and 60 reported another language as their mother tongue.

### 2.3. Questionnaire

The online questionnaire included five scales (in addition to a short section on demographics, see Table 1 above), each measuring concepts under study, i.e., quantity and quality intergroup contacts, knowledge sharing; attitudes about older workers; work engagement and intentions to remain. Participants completed the questionnaire according to their age group (e.g., 49 years old or less; 50 years old or more), in either French or English. Translation from English to French was carried out by researchers relying on a back-to-back translation process.

#### 2.3.1. Quantity and Quality Intergroup Contacts (QQIC)

Participants completed the 8-item *Intergroup Contacts Quantity and Contact Quality Scale* (ICQCQS) by Islam and Hewstone [41] that measures the frequency of contacts as well as the valence of the contact experience between younger and older workers. The scale comprises two factors, i.e., four items aim at assessing the quantity of contacts, for example: how much contact do you have with younger workers within your workplace (for participants aged 50 years old or more); how much contact do you have with older workers within your workplace (for participants aged 49 years old or less); and four items measuring the quality of contacts (e.g., to what extent did you experience the contact with younger workers as competitive or cooperative/to what extent did you experience the contact with older workers as competitive or cooperative). Quantity intergroup contact items were assessed using a Likert-scale ranging from 1 (never) to 7 (always) while quality intergroup contacts were measured with a Likert-scale ranging from 1 (definitely not) to 7 (definitely). Internal reliability of both subscales was acceptable with Cronbach’s alpha values of 0.72 and 0.84, respectively. The overall mean of the two subscales was used to calculate a composite score. Higher scores indicated more frequent and better quality age-based intergroup contact.

#### 2.3.2. Knowledge-Sharing Practices

In addition to the perception of quantity and quality intergroup contacts, the questionnaire also assessed the extent to which younger and older workers tend to share their knowledge (in a bidirectional mode, e.g., collecting and donating knowledge). To do so, participants completed the Knowledge Sharing Behavior Measure (KSBM), by Kim and Lee [42], comprising of eight items; four of these target collecting knowledge practices, for example: When I need certain knowledge, I ask my younger colleagues about it (for participants aged 50 years old or more); when I need certain knowledge, I ask my older colleagues about it (for participants aged 49 years old or less). Four other items focus on donating behaviors, for example: when I have learned something new, I tell my younger colleagues about it (for participants aged 50 years old or more); when I have learned something new, I tell my older colleagues about it (for participants aged 49 years old or less). Participants answered these statements using a Likert-scale ranging from 1 (completely disagree) to 7 (completely agree). Both subscales had an acceptable internal consistency with alpha values of 0.83 and 0.89, respectively, which is quite similar to the overall value of the original scale (α = 0.85). A composite score was calculated using the overall mean of the two subscales with higher scores indicating higher levels of knowledge (collecting and donating) practices between younger and older workers.

#### 2.3.3. Attitudes about Older Workers

Participants were asked to complete a 12-item scale measuring positive oriented attitudes about older workers, adapted from Warr and Pennington [43], by Iweins, Desmette and Yzerbyt [44]. The scale comprises two subscales (6 items each) measuring the level of effectiveness (e.g., older workers are conscientious); and adaptability of older workers (e.g., older workers learn quickly). Participants answered the statements on a Likert-scale ranging from 1 (completely disagree) to 7 (completely agree). Internal consistency alpha values of both subscales were 0.90 and 0.92, respectively, both values being in the range of the ones of the original scale (α values ranging from 0.87 to 0.95) [36,44]. A composite score was calculated based on the overall mean of the two subscales. A higher composite score indicated more positive attitudes towards older workers.

#### 2.3.4. Work Engagement

Participants were also asked to complete the Utrecht Work Engagement Scale (UWES), developed by Schaufeli, Salanova, Gonzales Romá and Bakker [45], measuring their level of engagement toward their work. Originally, this scale comprised 17 items scored according to three subscales. The three subscales aimed to measure the concepts of Vigor—3 items (e.g., I can continue working for very long periods at a time); Dedication—9 items (e.g., my job inspires me); and Absorption—5 items (e.g., I am immersed in my work). Participants responded to the statements according to a Likert-scale ranging from 1 (never) to 7 (always). Alpha values of the three subscales were 0.74, 0.95, 0.83, respectively; values that are similar to the overall one of the scale (α = 0.89) suggests good internal reliability. The overall mean score of the three subscales were used to calculate a composite score. A higher composite score on this scale was indicative of a higher level of engagement at work.

#### 2.3.5. Intentions to Remain

To assess the intention of workers to remain with their organization, participants completed a short scale developed by Armstrong-Stassen and Ursel [46]. This one-factor scale includes three items measuring intentions to remain (e.g., I expect to continue working as long as possible in this organization). Participants were asked to answer statements using a Likert-scale ranging from 1 (completely disagree) to 7 (completely agree). The alpha value of 0.91 (which is higher than the value of the original scale, α = 0.85) suggests that the scale has a very good internal reliability. A composite score was calculated by averaging the score on each of the three items. A higher score indicates stronger intentions to remain within the organization.

#### 2.3.6. Age

Participants’ age was chosen as a control variable following univariate analyses that showed significant correlations between Age and the dependent variables, but also because of previous findings. In fact, age has an inherent relationship with attitudes and stereotypes that can be observed about older workers [47,48]. It also influences the involvement that people show for their work [49], as well as the willingness they have to remain in their current organization [50]. As such, Age as a continuous variable, was used as a control variable in the hypothesized model. The continuous form of the variable was used to avoid any potential effect of the differences between the sample size of each age category. Thus, Age was used as control in the hypothesized model on the Attitudes About Older Workers, Work Engagement, and the Intention to Remain within their organization.

### 2.4. Data Analyses

Statistical analyses were conducted using Statistical Package for Social Sciences (IMB SPSS software, version 27) and IBM SPSS AMOS software (version 27). A four-step process was used: data screening and cleaning was first conducted, followed by confirmatory factor analysis (CFA) on each measurement scale; descriptive statistics and correlations were then performed, as well as path analyses.

#### 2.4.1. Screening and Missing Data

A total of 142 participants were discarded from the original sample based on demographics, missing data, and response invariance (calculated as an overall standard deviation across scale responses below 0.50). Participants aged less than 18 years old and those with a completion rate lower than 90% were discarded from the original sample. The final sample included 603 participants.

Patterns of missing data were not considered since the percentage of missing data for each scale was below 5% [51]. However, mean substitution was used on missing data to preserve consistency in the sample size across all analyses.

#### 2.4.2. Multivariate Assumptions

Data distribution and assumptions for multivariate analyses were assessed for all variables and the significance level for all analyses was set at *p* < 0.05. Outliers and influentials were examined first to determine whether our data contained multivariate outliers. To do so, we ran Cook’s distance analysis [52]. Since most cases had Cook’s distance below 1, with most of them being less than 0.10, we can assume that we did not have any influential outliers.

Regarding multicollinearity, variable inflation factors (VIF) were used for all predictors on both dependent variables (i.e., work engagement and intention to remain). Results showed that all VIF values were below 1.7; since those values are far below the threshold of 5 [53], we can confirm that this assumption was respected.

#### 2.4.3. Factor Analyses

The underlying structure of each scale was retrieved using CFA which led to the removal of one item in the Knowledge-Sharing Practices scale and some structural differences in subscales for the Work Engagement scale. Subsequently, descriptive statistics and correlations were performed (see Table 2) before assessing the postulated model using path analysis. Indicators of goodness of fit [54] were examined in CFA and in model testing with the following cut-off values: the Chi-Square (*p* > 0.05); the Chi-Square/Degrees of Freedom (CMIN/DF; <3), the Root Mean Square Error of Approximation (RMSEA; <0.05); the Comparative Fit Index (CFI; >0.90) and the Tucker–Lewis Index (TLI; >0.90).

#### 2.4.4. Model Selection and Control Variables

During model testing, links between the variables were added or removed from the hypothesized model using stepwise selection based on the fit indices provided by AMOS. The model was further inspected using Age as a control variable to isolate its potential effect and keep results stable independent of participant’s age.

## 3. Results

### 3.1. Descriptive Statistics and Correlations

Descriptive statistics including mean, standard deviation and intercorrelations for each scale and the control variable are presented in Table 2. Mean values for all scales ranged from 4.92 and 5.29 out of a possible total score of 7. The value of Skewness and Kurtosis varied from −0.73 to −0.26 and −0.47 to 0.72, respectively, and were all situated in satisfactory ranges suggesting normal distributions.

Correlations between the control variable and subscales revealed that participants’ age was positively correlated with all subscales (*r* values ranging from 0.11 to 0.39, *p* < 0.01). The intercorrelation matrix revealed that subscales were all positively correlated with each other with *r* values ranging from 0.16 and 0.77 (*p* < 0.01).

### 3.2. Path Analysis

The hypothesized model (see Figure 1), including age as a control variable on all dependent variables, was tested through path analysis. Indicators of goodness of fit indicated that the hypothesized model did not adequately fit the data with χ² (4) = 71.96, *p* < 0.001 (CMIN/DF = 17.99, TLI = 0.71, CFI = 0.92, RMSEA = 0.17).

Using a stepwise process, the goodness of fit was significantly improved by the addition of three connections and the removal of two postulated ones. As such, in the final model, the variable Quantity and Quality Intergroup Contacts has an effect on Work Engagement and Intention to Remain, while the variable Knowledge-Sharing Practices is connected to Work Engagement. However, the association between Attitudes About Older Workers and Intention to Remain and the control effect of Age on Work Engagement were excluded from the final model. Following these modifications, results indicated a good fit for the final model illustrated in Figure 2, χ² (3) = 5.09, *p* = 0.17 (CMIN/DF = 1.70, TLI = 0.98, CFI = 0.99, RMSEA = 0.03).

### 3.3. Effects in the Final Model

Overall, the final model supported four out of the five hypotheses, but failed to confirm the predicted association between the Attitudes About Older Workers and the Intention to Remain.

As predicted, Quantity and Quality Intergroup Contacts and Knowledge-Sharing Practices were both positively correlated with Attitudes About Older Workers (*β* = 0.25 and *β* = 0.26, *p* < 0.001, respectively). In addition, Attitudes About Older Workers was positively related with Work Engagement (*β* = 0.16, *p* < 0.001) which was itself positively correlated with the Intention to Remain (*β* = 0.44, *p* < 0.001). In regard to the associations that were added to the model, Quantity and Quality Intergroup Contacts was positively related with Work Engagement (*β* = 0.21, *p* < 0.001) and with the Intention to Remain (*β* = 0.10, *p* < 0.05), while Knowledge-Sharing Practices was positively associated with Work Engagement (*β* = 0.19, *p* < 0.001). 

As for the control variable, the positive relationship between *Age* and *Attitudes About Older Workers* was confirmed (*β* = 0.08, *p* < 0.05), as well as its positive link with the *Intention to Remain* (*β* = 0.08, *p* < 0.05). However, the control effect of *Age* on *Work Engagement* was not statistically significant (*β* = 0.06, *p* = 0.11). In other words, a greater age was associated with more positive attitudes about older workers and stronger intentions to remain within the organization, but it did not seem to affect engagement at work in this model.

## 4. Discussion

The goal of the current study was to examine factors that may contribute to reducing ageism in the Canadian workplace. Building on results from previous studies as well as the main postulates of the intergroup contact theory, it was hypothesized that a workplace thriving on quality and quantity intergenerational contacts and knowledge sharing between younger and older workers would generate positive attitudes regarding the latter; in turn, such attitudes would increase work engagement levels and intentions to remain with the organization. Five of the six hypotheses comprised in the initial model were confirmed. The final model also includes additional links that were not predicted, suggesting a direct impact of intergenerational contacts on work engagement and intentions to remain as well as of knowledge sharing on levels of work engagement. Each of these links is discussed in the following sections.

### 4.1. Indirect and Direct Effect of QQIC

As predicted, QQIC has a positive effect on the way older workers are perceived, which in turn, increases the level of work engagement. Such findings support the Intergroup Contact Hypothesis’s main postulate that bringing together members of different groups, in this case workers of different ages and generations, can reduce age-based prejudice, precisely ageist stereotypes. In explaining how intergroup contacts are efficient in prejudice reduction, Pettigrew and Tropp [34] sustain that intergroup interactions allow members to complete the information they may be missing about one another and as such avoid stereotype development. More so, as initially conceptualized by Allport [33], the effect of intergroup contacts on prejudice cannot be solely explained by the frequency of such contacts but also, and most importantly, by their quality; hence the combination of both types of contacts was used in the current study to measure their effect on positive attitudes toward older workers.

As predicted, the final model suggests that the more positive are the attitudes about older workers, the stronger is work engagement. These results fall in line with previous studies showing that perceived ageist attitudes and discrimination in the workplace is associated with decreased work engagement [55,56].

The final model also suggests two direct effects of QQIC that were not predicted initially, namely a positive effect on work engagement and intentions to remain with the organization. Importantly, these effects stand for participants of all ages. In other words, for both younger and older workers, intergroup contacts not only generate positive attitudes toward the latter but increase levels of work engagement and intentions to remain with the organization. Although not predicted, these findings shed further light on the intergroup contact theory, suggesting that increasing the frequency and the quality of contacts between younger and older workers is beneficial in terms of increased levels of engagement and intentions to stay with the organization. As stated above, in the context of major labor shortages, especially in Canada, the direct and positive link between the variables “Quantity and Quality Intergroup Contacts” and “Intentions to remain” paves the way for an interesting course of action for employers and managers to take.

The final model also confirms the hypothesis as per the impact of age-based knowledge-sharing practices (KSP) between younger and older workers on positive attitudes related to the latter. Knowledge sharing behaviors actually exemplify quantity and quality intergroup contacts, hence explain the significant and positive correlation between these two variables and their effect in reducing negative ageist stereotypes in the workplace. It is worth noting that knowledge sharing was measured through a bidirectional dimension in that both younger and older workers were asked the extent to which they donated but also collected knowledge from each other. Expanding the intergroup contact theory, these results reveal the importance of a work culture that supports knowledge sharing between both younger and older workers in order to foster positive attitudes about aging in the workplace. Similar to the predicted and confirmed link between QQIC and positive attitudes about older workers, the impact of knowledge sharing in bolstering positive views on older workers may be explained by a filling or correction mechanism, whereby the experience of sharing knowledge in itself allows workers to get to know each other beyond stereotypical views based on age.

The final model also reveals the crucial role that knowledge sharing plays not only in changing attitudes about older workers but also in boosting work engagement for both younger and older employees. Although not predicted and similar to the direct association between QQIT and work engagement, this result, which falls in continuity with previous studies [57,58], speaks to the importance of putting in place concrete workplace behavioral initiatives that allow for younger and older workers to interact with one another.

Results suggest that holding positive attitudes toward older workers does not have a direct effect on the intention to remain in the workplace. In fact, only QQIC and work engagement had direct effects on the intention to remain. This nonsignificant finding may be explained by the fact that attitudes are a psychological construct and may not necessarily be manifested in behavior [59]. Consequently, it is plausible to think that even if colleagues think highly of older workers’ effectiveness and adaptability, these thoughts and feelings may not be sufficient to influence one’s intention to stay in the workplace. On the other hand, perceived behavioral evidence such as positive intergroup contacts and work engagement through vigor and absorption are concrete, more easily recognized and validated, which can explain their direct impact on one’s intention to remain with the organization.

From a practical perspective, these findings suggest that efforts can be made to modernize workplace climate, policies and practices by focusing on improving the quantity and quality of intergroup contacts as well as enhancing knowledge-sharing practices between different age groups. Employers are encouraged to invest in evidence-based strategies that promote intergenerational interactions [60]. Organizational policies can be implemented to promote the intentional building of intergenerational teams for specific workplace initiatives, such as fundraising campaigns or social events, to enhance frequency and quality of intergroup contacts [61,62]. In-person or online intergenerational programs, such as intergenerational and reverse mentoring as well as intergenerational training and workshops—activities that promote the exchange of knowledge, skills, and experience between younger and older workers—can be established in the workplace to help increase both intergroup contacts and knowledge sharing across age groups [62,63,64]. Employers can also encourage participation in workplace health and wellness programs through fitness classes or lunch walks as an age-friendly workplace opportunity where employees of all ages can informally socialize and develop organizational social support networks across generations [65,66].

### 4.2. Limitations

The current study is not without limitations, namely, three are important to state. First, the study’s cross-sectional nature prevented the examination of causal relationships over time. Second, while both quantity and quality of intergroup contacts were measured, we did not specifically examine the four optimal conditions of intergroup contact (a combination of high/low frequency and quality) which may have resulted in different effects on the study variables. Third, considering that the study data were collected during the COVID-19 pandemic where work environments were rapidly changing, it is not clear whether the ICQCQS effectively assessed in-person, virtual, and other forms of intergroup contacts. Finally, although the age of participants ranged from 18 to 68 years old, the mean age was 37.58 years old with a standard deviation value of 11.53. Hence, intergroup contact was tested on a rather homogenous group of participants regarding age. Future studies should ensure heterogeneity based on age within and between categories. Such future studies should also consider variables such as ethnicity and cultural beliefs of workers, as it is plausible to think that they may play a role in terms of intergenerational contacts and attitudes about aging.

## 5. Conclusions

Despite these limitations, this study makes a theoretical contribution to the scientific literature by advancing existing knowledge of the intergroup contact theory (ICT). Relying on the postulates of this theory, the final model highlights the central role played by quality and quantity intergroup contact in reducing age-based prejudice and more so, in increasing work engagement and intentions to remain for both younger and older workers. As such, the study further develops understanding of ageism in the Canadian labor market, and provides scientific evidence in support of employment policies and practices that foster inclusive and respectful multigenerational workplaces. Studies such as this one—where intergenerational contact helps to combat ageism—provide hope for the future of work.

## Figures and Tables

**Figure 1 ijerph-19-04866-f001:**
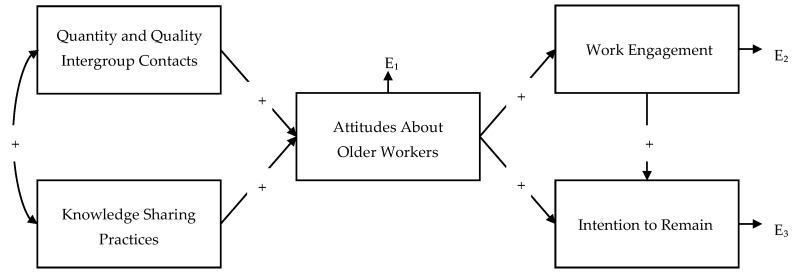
Path diagram of the hypothesized model. Control variables are omitted in the figure. In this model, age controls for Attitudes About Older Workers, Work Engagement, and Intentions to Remain.

**Figure 2 ijerph-19-04866-f002:**
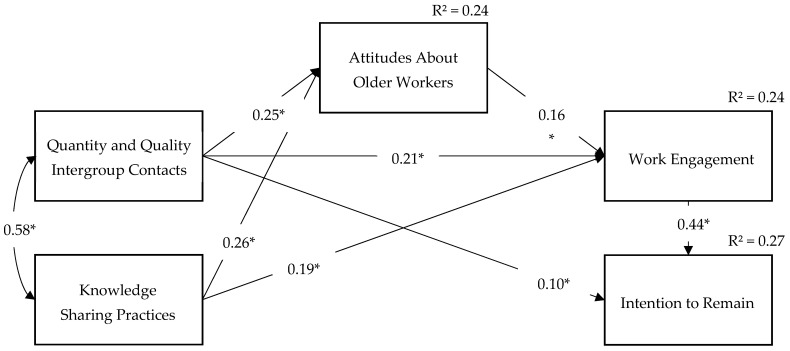
Final path diagram and hypotheses testing. Control variables are omitted in the figure. *Model fit.* χ² (3) = 5.09, *p* = 0.17 (CMIN/DF = 1.70; TLI = 0.98; CFI = 0.99; RMSEA = 0.03). *Notes.* RMSEA = root mean square error of approximation; CFI = comparative fit index; TLI = Tucker–Lewis Index. In this model, age acts as a control variable on Attitudes About Older Workers and Intentions to Remain. * *p* < 0.05.

**Table 1 ijerph-19-04866-t001:** Frequencies per Demographic Characteristics of the Sample.

	Age Categories (Years Old)	
	18–49 (*n* = 489)	50–68 (*n* = 114)
Gender		
Female	266	74
Male	223	40
Province		
Quebec	252	91
Ontario	112	12
Other	125	11
First language		
French	230	84
English	208	21
Other	51	9
Hours worked per week		
20 h or less	50	7
More than 20 h	439	107
Manager		
Yes	120	36
No	369	78
Work sector		
Private	227	39
Public	262	75

Notes. Missing data are not included in the table.

**Table 2 ijerph-19-04866-t002:** Means, Standard Deviations and Intercorrelations Matrix for All Subscales and Control Variable (N = 603).

Measure	M (SD)	1	2	3	4	5	6	7	8	9	10	11
1. Age	37.58 (11.53)	–										
2. Contact Quantity	4.61 (1.19)	0.39 *	–									
3. Contact Quality	5.47 (1.06)	0.32 *	0.39 *	–								
4. Knowledge Collecting	5.29 (1.14)	0.11 *	0.29 *	0.36 *	–							
5. Knowledge Donating	5.28 (1.24)	0.38 *	0.50 *	0.48 *	0.51 *	–						
6. Stereotypes Bias	5.37 (.97)	0.16 *	0.21 *	0.41 *	0.35 *	0.32 *	–					
7. Adaptability	4.32 (1.26)	0.30 *	0.28 *	0.41*	0.32 *	0.36 *	0.61 *	–				
8. Vigor	5.25 (1.01)	0.19 *	0.23 *	0.32*	0.25*	0.31 *	0.28 *	0.21 *	–			
9. Dedication	5.02 (1.12)	0.29 *	0.32 *	0.40*	0.26*	0.41 *	0.30 *	0.30 *	0.63 *	–		
10. Absorption	4.55 (1.08)	0.16 *	0.24 *	0.30*	0.24*	0.29 *	0.27 *	0.30 *	0.58 *	0.77 *	–	
11. Turnover	4.94 (1.70)	0.23 *	0.23 *	0.31 *	0.16 *	0.21 *	0.18 *	0.23 *	0.28 *	0.52 *	0.44 *	–

Notes. For all scales, higher scores indicate more extreme responding in the direction of the construct assessed. Turnover = intentions to remain within the organization. * *p* < 0.01.

## Data Availability

The data presented in this study is available from the respective author upon request.

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
