# Peer review of "Testing the Shielding Effect of Intergenerational Contact against Ageism in the Workplace: A Canadian Study"

_ijerph, 2022, doi:10.3390/ijerph19084866_

Round 1

Reviewer 1 Report

It was a pleasure reviewing your well-organized and well-written manuscript. I think it contributes to the literature on ageism in the workplace and Intergroup Contact Theory. 

I have a few comments below that could be addressed to strengthen the links between theory and the hypotheses and clarify your discussion about age categories.  

I think it would be helpful to have a bit more detail about Intergroup Contact Theory, specifically the four optimal conditions, before line 123 ("Interestingly, Pettigrew & Tropp..."). This would ensure that readers who may not be familiar with this theory have sufficient information to contextualize the subsequent literature review. On line 154, you refer to "...building on the findings from previous studies stated before..." but how about theory (i.e., as noted on lines 379-380 "Building on results from previous studies as well as the main postulated of the Intergroup Contact Theory, it was hypothesized that...")? I was also curious about how the four optimal conditions informed (or don't) the hypotheses and discussion section or is the link at a higher level (i.e., intergenerational contact regardless of the presence of these conditions). You discuss this in the limitations section (lines 470-471) but clarifying this prior to the statement of hypotheses was needed.

Did it matter that there were 489 participants aged 18-49 vs only 114 aged 50-68? This could be specifically mentioned in the results section.  

In the limitations section, it would be beneficial to point out concerns about age homogeneity. You mention the potential for having a lower age  for "older workers" and "Using different age categories would have potentially influenced the results." It wasn't clear whether your reference to "different age categories" referred to simply having a lower age for "older workers" or that future research could consider more categories of age of both older and younger workers (e.g., 18-29, 30-39, 40-49, 50-59, 60-69). Additional categories would counter homogeneity which contributes to stereotypes and ageism. This could be discussed further. 

In the conclusion you mention that "...this study makes a theoretical contribution to the scientific literature by advancing existing knowledge of intergroup contact theory" (line 480-481). I suggest specifying in which ways it was advanced. This goes back to my previous comment above about how theory informed the hypotheses. 

Author Response

Reviewer 1: 

I think it would be helpful to have a bit more detail about Intergroup Contact Theory, specifically the four optimal conditions, before line 123 ("Interestingly, Pettigrew & Tropp..."). This would ensure that readers who may not be familiar with this theory have sufficient information to contextualize the subsequent literature review.

Response: Thank you for this suggestion. We have added the following text starting at line 123: The first condition relates to how groups expect and perceive equal status in the situation while the second, common goals, is about contact that requires active and goal-oriented effort. Athletic teams are an example of this second condition. Intergroup cooperation relates to the interdependent nature of the efforts needed to achieve the common goals but without intergroup competition. Finally, the fourth condition states that intergroup contact is more easily accepted when explicitly support by authorities.

On line 154, you refer to "...building on the findings from previous studies stated before..." but how about theory (i.e., as noted on lines 379-380 "Building on results from previous studies as well as the main postulated of the Intergroup Contact Theory, it was hypothesized that...")?

Response: Thank you for this important point. The study relies entirely on the ICT in predicting that intergroup contacts can reduce age based prejudice in the workplace (and as such, measures the impact of such contacts). Within the manuscript, we have modified the first sentence under 1.3, starting on line 160 to: Hence, building on the postulates of the ICT as well as findings from previous studies stated above, the current study aims at assessing…

I was also curious about how the four optimal conditions informed (or don't) the hypotheses and discussion section or is the link at a higher level (i.e., intergenerational contact regardless of the presence of these conditions). You discuss this in the limitations section (lines 470-471) but clarifying this prior to the statement of hypotheses was needed.

Response: In light of Pettigrew’s meta-analysis suggesting that the four conditions were not needed for intergroup contacts to significantly reduce prejudice, these conditions were not included in the model. However, we fully agree with the reviewer that this is a limitation of the study, as we have stated in the limitation section.

Did it matter that there were 489 participants aged 18-49 vs only 114 aged 50-68? This could be specifically mentioned in the results section.  

Response: The difference in the number of participants in each age category did not impact the model since age was controlled for and used as a continuous rather than categorial variable in the model itself. Please note that further details have been added in Section 2.3.6 Age, starting on line 280.

In the limitations section, it would be beneficial to point out concerns about age homogeneity. You mention the potential for having a lower age  for "older workers" and "Using different age categories would have potentially influenced the results." It wasn't clear whether your reference to "different age categories" referred to simply having a lower age for "older workers" or that future research could consider more categories of age of both older and younger workers (e.g., 18-29, 30-39, 40-49, 50-59, 60-69). Additional categories would counter homogeneity which contributes to stereotypes and ageism. This could be discussed further.

Response: This was addressed in the revised manuscript starting on line 482: Finally, although the age of participants ranged from 18 to 68 years old, mean age was 37.58 years old with a standard deviation value of 11.53. Hence, intergroup contact was tested on a rather homogenous group of participants as it relates to age. Future studies should ensure heterogeneity based on age within and between categories.

In the conclusion you mention that "...this study makes a theoretical contribution to the scientific literature by advancing existing knowledge of intergroup contact theory" (line 480-481).

Response: Thank you for this comment. We have clarified how this study advances the intergroup contact theory, starting on line 491 of the Conclusions section.

Reviewer 2 Report

Thanks for inviting me to review "Testing the shielding effect of intergenerational contact against ageism in the workplace: a Canadian study".
Overall the manuscript reads very smoothly, with a fluent introduction, literature, and discussion. However, some issues appeared in the methodological part, which might be very critical.

1, did not report the internal consistency of subscales. Usually, we only do composite score computation of a subscale after a reliability check (internal consistency) of this subscale.
2, did not include measurement of demographic characteristics, such as age. From table 1, it looks like a categorical variable, but in the results, it read a continuous one. descriptions of measurement, including demographics, should be added. Also, if age is measured by years of age, its means and stand errors should be present in table 1.
Line 309-316 should move to the measurement part.
3, CFA. In lines 296-298, it was not necessarily of using CFA for well-developed scales. Reporting alphas (of subscales) is good enough. If the alphas meet the standard, item deletion is not needed, as they were very well-developed and only a result within one sample is not persuasive to change the indicators (items) for a concept.
4, In SEM, CFI and TLI >1 are usually ill-fitted. 1.00 is also partially indicating that the model is possibly just-identified. So please provide the exact model fit indexed with two decimal places. If the results do not look good, a personal suggestion is to use the latent model for instruments with subscales, and you can also possibly consider the packing technique.

Some other comments:
1, please move Figure 1 to place it near your hypotheses, which would be very friendly to read.
2, line 46 seems should be "Harris et al.,".
3, use two decimal places consistently throughout the manuscript, including model-fit index and p-value reporting.
4, table 2, "N = 603" should be put at the end of the table header with a bracket. As you conducted analyses based on subscales, all subscales should be presented here.
5, please remove “age” from the figures and give a note "control variables, and error terms are omitted in the figure".
6, please also report chi-squared/df in SEM model fit (threshold is <3).

Author Response

Reviewer 2: 

1, did not report the internal consistency of subscales. Usually, we only do composite score computation of a subscale after a reliability check (internal consistency) of this subscale.

ResponseWe thank the reviewer for this comment. The values were checked before computing the composite scores but were not included in the first version of the manuscript. Following this comment, the values for each subscale have been added in the revised version.  Please review Section 2.3 Questionnaire under the Materials and Methods section, starting on line 204.

2, did not include measurement of demographic characteristics, such as age. From table 1, it looks like a categorical variable, but in the results, it read a continuous one. descriptions of measurement, including demographics, should be added. Also, if age is measured by years of age, its means and stand errors should be present in table 1

Response: In the survey questionnaire, age was measured as a continuous variable while age categories were created during data analysis (in order to distinguish between younger workers – 18 to 49 years old  - and older workers – 50 to 68 years old  – ), as reported in Table 1 (Frequencies per Demographic Characteristics of the Sample). However, the hypothesized model was tested with age as a continuous variable. Its mean and standard deviation values are reported along other continuous variables, in Table 2.

Line 309-316 should move to the measurement part.

Response: This paragraph has been moved to section 2.3.6, starting on line 280.

CFA. In lines 296-298, it was not necessarily of using CFA for well-developed scales. Reporting alphas (of subscales) is good enough. If the alphas meet the standard, item deletion is not needed, as they were very well-developed and only a result within one sample is not persuasive to change the indicators (items) for a concept.

Response: We thank the reviewer for these comments. Although the scales in our study were already developed, we decided to conduct CFA to validate the scales with this specific population. More so, relying solely on internal reliability values does not allow to determine the latent structure of a scale and as such may generate errors of interpretation. In the case of the Knowledge Sharing Behavior scale, the alpha value was not satisfying when all items were included. Following CFA, one item of that same scale had to be removed (i.e. item no. 4:  When one of my younger colleagues is good at something, I ask him/her to teach me how to do it) to actually increase internal reliability. It is important to note that this same item was also removed from another sample in a study conducted by Lagacé, Van de Beeck and Firzly (2019). More generally, results from CFA can be useful to report to researchers in case they encounter issues when retrieving the structure of the scales from different populations/samples.  

In SEM, CFI and TLI >1 are usually ill-fitted. 1.00 is also partially indicating that the model is possibly just-identified. So please provide the exact model fit indexed with two decimal places. If the results do not look good, a personal suggestion is to use the latent model for instruments with subscales, and you can also possibly consider the packing technique.

Response: Thank you for this valuable comment. We have provided the exact model fit indexed with two decimal places. Given that the values for CFI and TLI were Ë‚ 1 (.98 and .99 respectively), the model is not just-identified. Modifications can be seen in the notes of Figure 2 and in Section 3.2 Path Analysis.

please move Figure 1 to place it near your hypotheses, which would be very friendly to read.

Response: We have made this change.

line 46 seems should be "Harris et al.,". 

Response: We have made this change.

Use two decimal places consistently throughout the manuscript, including model-fit index and p-value reporting

Response: Decimals have been adjusted throughout the manuscript.

Table 2, "N = 603" should be put at the end of the table header with a bracket. As you conducted analyses based on subscales, all subscales should be presented here

Response: Thank you for these suggestions. Sample size value has been moved to the header with a bracket. All subscale values are now presented in Table 2.

Please remove “age” from the figures and give a note "control variables, and error terms are omitted in the figure"

Response: These modifications have been made; see Figure 1 and Figure 2.

Please also report chi-squared/df in SEM model fit (threshold is <3).

Response: Thank you for this suggestion. The chi-squared/df (CMIN/DF) have been added in the cut-off values (see Lines 324 – 325), as well as in the model fit of the hypothesized model (see Lines 347 – 348) and the final model (see Lines 356 – 357) and in the notes of Figure 2.

Reviewer 3 Report

Strengths: Literature review, methodology, sample, results

Weaknesses: nothing to report

Improvement proposals

Line 173 – “After obtaining ethics approval from the first author’s institution, the team of researchers recruited participants through online invitations.” Authors should mention the criteria for inviting participants in this study.

Line 478 - Using different age categories would have potentially influenced the results. Authors must substantiate this idea.

Line 485 - Studies such as this one – where intergenerational contact helps to combat ageism provide hope for the future of work.” Authors should describe what kind of future studies may follow this one.

Author Response

Reviewer 3: 

Line 173 – “After obtaining ethics approval from the first author’s institution, the team of researchers recruited participants through online invitations.” Authors should mention the criteria for inviting participants in this study.

Response: We added the inclusion criteria to this paragraph under Section 2.1 Recruitment, starting on line 190.

Line 478 - Using different age categories would have potentially influenced the results. Authors must substantiate this idea.

Response: Thank you for this comment. We have modified this part of the Limitations section to further substantiate this idea. The following has been added starting on line 482: Finally, although the age of participants ranged from 18 to 68 years old, mean age was 37.58 years old with a standard deviation value of 11.53. Hence, intergroup contact was tested on a rather homogenous group of participants as relates to age. Future studies should ensure heterogeneity based on age within and between categories.

Line 485 - Studies such as this one – where intergenerational contact helps to combat ageism provide hope for the future of work.” Authors should describe what kind of future studies may follow this one.

Response: In our revised limitations section, we propose several future studies that could follow the current study, including those that ensure heterogeneity based on age within and between categories, as well as studies that consider variables such as ethnicity and cultural beliefs of workers. Please see lines 485 – 488.

Round 2

Reviewer 2 Report

Significant revisions have been made, and the current manuscript is almost ready for publish.

Please note that the "Control variables are omitted in the figure. In this model, age controls for Attitudes About Older Workers, Work Engagement, and Intentions to remain" should serve as a note for the Figure 1.

Author Response

Reviewer 2: 

Please note that the "Control variables are omitted in the figure. In this model, age controls for Attitudes About Older Workers, Work Engagement, and Intentions to remain" should serve as a note for the Figure 1.

Response: thank you to the reviewer for this comment. The note was already included under Figure 1 in the manuscript. We have also added the note to the attached document entitled: All tables and figures.